# Enhance Production of γ-Aminobutyric Acid (GABA) and Improve the Function of Fermented Quinoa by Cold Stress

**DOI:** 10.3390/foods11233908

**Published:** 2022-12-04

**Authors:** Yucui Zhang, Ming Zhang, Ting Li, Xinxia Zhang, Li Wang

**Affiliations:** 1School of Food science and Technology, Jiangnan University, Lihu Road 1800, Wuxi 214122, China; 2National Engineering Research Center of Cereal Fermentation and Food Biomanufacturing, Jiangnan University, Lihu Road 1800, Wuxi 214122, China; 3Jiangsu Provincial Engineering Research Center for Bioactive Product Processing, Jiangnan University, Lihu Road 1800, Wuxi 214122, China; 4Key Laboratory of Carbohydrate Chemistry and Biotechnology Ministry of Education, Jiangnan University, Lihu Road 1800, Wuxi 214122, China; 5Collaborative Innovation Center of Food Safety and Quality Control in Jiangsu Province, Jiangnan University, Lihu Road 1800, Wuxi 214122, China

**Keywords:** GABA, fermentation, cold stress, glutamate decarboxylase

## Abstract

Quinoa is an excellent source of γ-aminobutyric acid (GABA), which is a natural four-carbon non-protein amino acid with great health benefits. In this study, the quinoa was treated by cold stress before fermentation with *Lactobacillus plantarum* to enhance the amount of GABA. The best *Lactobacillus plantarum* for GABA production was selected from sixteen different strains based on the levels of GABA production and the activity of glutamic acid decarboxylase (GAD). Cold stress treatments at 4 °C and at −20 °C enhanced the amount of GABA in the fermented quinoa by a maximum of 1191% and 774%, respectively. The surface of the fermented quinoa flour treated by cold stress showed more pinholes, mucus, faults and cracks. A Fourier transform infrared spectrophotometer (FTIR) analysis revealed that cold stress had a violent breakage effect on the -OH bonds in quinoa and delayed the destruction of protein during fermentation. In addition, the results from the rapid visco analyzer (RVA) showed that the cold stress reduced the peak viscosity of quinoa flour. Overall, the cold stress treatment is a promising method for making fermented quinoa a functional food by enhancing the production of bioactive ingredients.

## 1. Introduction

Quinoa (*Chenopodium quinoa Willd*) is a pseudocereal that originates in the Andean region and has high resistance to various abiotic stresses, such as frost and hypoxia [1,2]. It is a grain that exhibits excellent nutritional values, attributable to its remarkable protein concentration (16.5−23.9%) and balanced amino acid composition, making it different from common cereals, such as barley and rice [3]. It is also a good source of beneficial micronutrients such as unsaturated fatty acids, vitamins, minerals and phenolic compounds [1]. Accordingly, quinoa has been recognized as a nutritive and healthy food and has inspired keen interest among researchers.

Various technologies have been employed to develop diverse quinoa products for the purpose of improving the diversity of quinoa consumption. In terms of feasibility and cost performance, fermentation is a valid method for enhancing the nutritional value and phytochemical content of quinoa [4]. It increases the digestibility of protein and starch, improves phenolic content and mineral bioavailability, and decreases some non-nutritive factors [5]. Several studies have shown that the fermentation of quinoa can spur physiological metabolic activity and promote the production of gamma(γ)-aminobutyric acid (GABA) [5,6], a four-carbon non-protein amino acid compound which is poorly represented in natural animal and plant foods [7]. It has been demonstrated that GABA may reduce anxiety and pain and lower blood pressure [2]. GABA also can activate the GABA_A_ receptor and GABA_B_ receptor in islet β-cells and thereby increase the release of insulin [8].

Generally, fermentation microorganisms catalyze the decarboxylation of L-glutamate by the glutamate decarboxylase (GAD) to synthesize GABA [7]. Furthermore, lactic acid bacteria (LAB) are regarded as safe and highly efficient microorganisms to use in manufacturing GABA-enriched fermented foods [7]. Accordingly, there are numerous studies about the selection or genetic modification of LAB with the aim of increasing GABA production by fermentation [9]. However, newly isolated LAB strains may be difficult to utilize broadly in commercial production, and the genetically modified strains may increase the safety concerns of some groups of people. As the synthesis of GABA is stimulated by diverse environmental factors [3], using physical methods to improve the GABA content of grains could be a good choice in order to avoid these issues.

Several physical methods have been used to enhance GABA production in fermented products, including hypoxia, cold stimulation, high pressure treatment, mechanical manipulation, pulsed electric fields and ultrasound [10]. Among them, cold stress stimulation in particular could be an excellent way to elevate the accumulation of GABA [11]. In a previous study, cold stress treatment resulted in a two-fold increasement of GABA synthesis in *Asparagus sprengeri* (Regel) mesophyll cells, which is attributed to the rapid and transient increasement of the cytosolic Ca^2+^ level [12]. However, information on the combination of cold stress treatment with fermentation to enrich GABA in cereals or pseudocereals is limited. To our knowledge, there is no report on the effect of cold stress pretreatment on GABA enrichment in fermented quinoa. The mechanism underlying the changes in the edible and functional properties of fermented quinoa after cold stress are still far from being clarified. In this context, the aim of this study is to maximize the GABA content in quinoa flour using a cold stress treatment and to investigate the alterations of starch crystallinity, the secondary structures of proteins and the enzymatic activity. Furthermore, the swelling as well as the pasting characteristics of the fermented quinoa flour are comparatively characterized. This work will promote the utilization of cold stress in the production of GABA-enrich fermented quinoa and provide foundational data for conditioning the end-use quality, as well as contributing to the development of novel bioactive pseudocereal products.

## 2. Materials and Methods

### 2.1. Materials

Quinoa (*Chenopodium quinoa Willd*) was provided by the Gansu Academy of Agriculture Science (Lanzhou, China). *Lactobacillus plantarum* NO.1−16 were isolated from human intestinal flora. GABA, MRS agar, and Monosodium L-glutamate (MSG) were purchased from Sigma Aldrich, Inc. (Shanghai, China). All chemicals are analytical grade unless otherwise stated.

### 2.2. Screening of GABA-Producing Strains 

*L. plantarum* was initially fostered in MRS Medium at 37 °C for 12 h in order to activate the strain, and then a 100 μL spore suspension was moved to the quinoa substrates (2.50 g) in an attempt to produce higher levels of GABA. Fermentation at 34 °C for 48 h was conducted to search for *L. plantarum* that produced high GABA levels when inoculated into the quinoa fermentation medium. A supernatant was obtained by centrifuging the fermentation liquid at 15,000× *g* for 0.5 h and analyzing the GABA content. 

### 2.3. Cold Stimulation Treatment and Fermentation Experiment

The quinoa seeds were disinfected with sodium hypochlorite (1% *w/v*) for 0.5 h and then cleaned with distilled water to a neutral pH. The quinoa was immersed in distilled water at room temperature for 1 d and then bleached twice with distilled water. The quinoa seeds were exposed to cold stress treatment at 4 °C and −20 °C for 24 h. The quinoa seeds were freeze-dried and ground through a 0.15 mm sieve to obtain quinoa flour. These samples were named Native, CQ−4 and CQ−20, respectively. After being pretreated, these samples were fermented for 0, 12, 24, 36, 48, 60, and 72 h using Incubator (Boxun, Shanghai, China), according to the following parameters: fermentation temperature 34 °C, solid-liquid ratio: 1:1.5.

### 2.4. Assessment of GABA and Glutamate 

GABA and glutamate were determined by the method developed by Baranzelli et al. [13], with some modifications. The samples (1 g) were added to trichloroacetic acid (10%, 25 mL) and left for over 2 h, then centrifuged for 30 min at 15,000× *g*. The supernatants were filtered through a 0.22 μm membrane and injected into HPLC vials for determination. GABA was measured by an Agilent 1100 liquid chromatograph on an Agilent Hypersil ODS column (250 mm × 4.6 mm) with a UV detector (at 338 nm) operating for 0−28 min at a column temperature of 40 °C. The flow phase was prepared using sodium acetate buffer (8 mg/mL, pH = 7.20 ± 0.02 and 220 μL of triethylamine, 5 mL of Tetralin (A) and sodium acetate buffer (8 mg/mL, pH = 7.20 ± 0.02), methanol, acetonitrile (1:2:2) (B). Two mobile phases flowed at 1.0 mL/min. 

### 2.5. Determination of pH, Total Acidity and Viable Microbial Counts

The total acidity and pH were determined at 0, 12, 24, 36, 48, 60, and 72 h of fermentation. Measurements were performed in triplicate according to the method described by Mahnoor et al. [14]. In brief, 2 g of each sample (Native, CQ−4, and CQ−20) were suspended in 18 mL of deionized water and stirred well under a stirrer (RW20 digital, IKA, Germany), and then left to stand for 10 min. The pH values were recorded using a pH meter (ST2100, LTD., Changzhou, China). The total acidity was determined by adding phenolphthalein as an indicator and dropping the sample’s supernatant (50 mL) with 0.1 M of sodium hydroxide, which showed a faint pink color that did not dissipate over 30 s. Then, 1 g each of the native, CQ−4, and CQ−20 samples were poured into a sterile test tube including 9 mL of sterile saline water and then vortexed. The different samples were diluted in a 10-fold gradient, and the dilutions were superficially plated on MRS agar for lactic acid bacteria in count. The Petri dishes were incubated (BXS-250S, Shanghai, China) at 37 °C for 48 h for the measurement of the total bacteria number [4].

### 2.6. Measure of Enzyme Activity

The measure of the GAD activity was performed using the method of Park et al. [15]. The samples (3 g) were mixed with 5 mL of 0.2 M sodium phosphate buffer (pH 5.7) and centrifuged at 8000× *g* for 10 min at 4 °C. The supernatant (1 mL) was incubated with 2 mL of glutamate solution (0.1%) at 40 °C for 2 h. The concentration of GABA in the reaction solution was measured by HPLC after the reaction was terminated by heating at 90 °C for 5 min. Each unit (U) of GAD was considered to be the number of units of the enzyme that form 1 μmol of GABA per minute [15].

### 2.7. FTIR

Variation in the secondary structure of the sample proteins as well as the structure of GABA was measured using a Fourier transform infrared spectrophotometer (FTIR, Nicolet, Madison, WI, USA) in the region from 400 to 4000 cm^−1^ with a resolution of 4 cm^−1^ and 32 scans. The samples were put on the ATR crystal for examination. When recording the spectra, the parameters were the same as the background. The Amide I band was used to analyze the protein’s secondary structure (1600−1700 cm^−1^). The crystalline structure of the quinoa starch was studied using R_1047_/_1022_ to express the starch’s crystallinity [2].

### 2.8. Crystallinity X-ray Diffraction (XRD)

The crystalline structure of the samples was analyzed in triplicate using an X-ray diffractometer (D2 PHASER, Bruker AXS, Germany) with a Cu Kα X-ray source operating at 40 kV and 30 mA. Samples with 7.0% moisture content were positioned evenly on a glass surface and scanned in the 2*θ* range of 4−40° with an increment of 1.2°/min. The relative crystallinity was calculated according to the method previously given by Li et al. [16].

### 2.9. Pasting Properties

The pasting properties were measured by a Rapid Visco Analyzer (Perten, Macquarie Park, NSW, Australia), following the AACC International Method 76-21.01. 

### 2.10. Scanning Electron Microscopy

The morphological characteristics of the samples (Native, CQ−4, and CQ−20) were viewed using a scanning electron microscope (SU8100, Tokyo, Japan) operated at 15 kV. These samples, which were fully dried at low pressure, were fitted to aluminum tubes with double-sided tape, and the surfaces were then sputtered with gold. Observations were made under a microscope with a magnification of 4000×.

### 2.11. Water-Soluble Index (WSI), and Water Absorption Index (WAI)

The WSI and WAI were measured according to the method of Qi et al. [17] with slight modifications. The sample (1.0 g) was dissolved in 50 mL of distilled water, agitated for 30 min in a water bath with a constant temperature (25 °C), and then chilled in an ice bath before centrifugation (4000× *g*, 10 min). The supernatant was moved to test tubes with a fixed weight. The tubes were then dried to a constant weight in a blast dryer at 105 °C. The original weight per unit of dry solids (WAI) was the weight of the gel after the supernatant has been removed. The weight of the dry solids in the supernatant was the WSI.

### 2.12. Statistical Analysis

All experiments were repeated three times, and data are presented as the mean of three samples ± standard deviation (SD). Using SPSS version 26.0 (SPSS Inc., Chicago, IL, USA), analysis of variance (*p* < 0.05) was used to evaluate statistical significance, followed by Duncan’s multiple range test. One-way analysis of variance (ANOVA) was applied to determine the significant differences of multiple variables (*p* < 0.05). Graphs were plotted using Origin 2018.

## 3. Results and Discussion

### 3.1. Screening of GABA-Producing LAB

During the current study, sixteen progressive strains were acquired by primary screening and then grown in quinoa grain medium and analyzed for their GABA production using HPLC (Figure 1). The strain named *L. plantarum* NO.2 had the highest GABA-producing ability. Its GABA content was 74.53 mg/100 g after 48 h in quinoa fermentation medium at 34 °C. The next highest producers of GABA were *L. plantarum* NO.4 and *L. plantarum* NO.3. Studies have suggested that the concentration of GABA in fermented brown rice is 22.91 μg/mL [3]. Therefore, *L. plantarum* NO.2 was established to have a higher ability to produce GABA. Furthermore, GABA production was induced by the direct decarboxylation of L-glutamate catalyzed by GAD, and GAD activity was essential for the production of GABA by LAB [18]. Therefore, further research was needed to confirm whether the strains with high GABA production also had high GAD enzyme activity. To exclude the interference of quinoa factors, strains cultured in MRS medium were chosen to determine the GAD enzyme activity. It was found that *L. plantarum* NO.2 still showed high GAD activity. The next highest exhibitors of GAD enzyme activity were *L. plantarum* NO.4 and NO.3. As a result, *L. plantarum* NO.2 was considered as a potential strain for high GABA production. 

### 3.2. Impact of Pretreatment and Fermentation on GABA Level of Quinoa

The GABA levels of quinoa flour from cold stress under 4 °C and −20 °C conditions are shown in Figure 2A. Comparing the different fermentation times, a maximum increment of 26.9% was observed in the native quinoa flour fermented for 48 h. The GABA content of the CQ−4 and CQ−20 samples also peaked at the fermentation time of 48 h. This indicates that fermentation time has the same effect on quinoa flour. Yang et al. also found that the optimal condition for GABA fermentation was 48 h. *Lactobacillus brevis* HYE1 was reported to produce about 14.64 mM GABA at 48 h of fermentation [19]. In the presence of glutamic acid decarboxylase (GAD), the main cause of GABA production is L-glutamic acid decarboxylation [2]. As the fermentation time continued, the GABA content began to decrease. This may be due to the depletion of fermentation substrates and the consumption of GABA by lactic acid during the fermentation process, or by the protein network formed by the binding of GABA with free amino acids [20]. Cold stress processing is a powerful method for accumulating GABA in fermented cereals. GABA content was enhanced by cold stress compared to the native quinoa flour. Moreover, compared to −20 °C cold stressed quinoa flour and native quinoa flour, the 4 °C cold stress treatment resulted in higher levels of GABA, which was attributed to lower GAD enzyme activity (Figure 3) and less glutamate consumption (Figure 2B). The GABA content of cold-stress-treated quinoa at 4 °C and 20 °C was 460% and 418% higher, respectively, than that of the untreated quinoa. The increase in GABA under cold stress may be attributed to an adaptive response of the cells to cold, which leads to acidification of the cell membrane. This is due to the depletion of H+ caused by decarboxylation during the synthesis of GABA. At the same time, the influence of cold stress on GABA accumulation was due to alterations in some key molecules of glutamate or GAD activity [21]. Ewa et al. demonstrated that cold stress formatted GABA was attributed to elevated Ca^2+^ levels in the cytosolic membrane. In addition, cold stress improved the soluble sugars of the cereal, and the elevated soluble sugars would be used by the strain as an available carbon source. Furthermore, the cold stress at 4 °C reached a maximum enhancement of 1191% compared to native quinoa flour after 48 h of fermentation (Figure 2A). The GABA content of the quinoa was enhanced by the pretreatment combined with fermentation, which means that a good synergy was produced by the cold stress treatment and fermentation. Not only that, but GABA can prevent the degradation of proteins and chilling damage [22]. Quinoa is a pseudocereal that can withstand the cold, and it raises its GABA content as part of its cold stress reaction. As a result of its high GABA content, quinoa is resistant to cold. Furthermore, because it does less damage to the nutritional composition, like gluten and starch, fermentation after cold stress may be advantageous for maintaining the function of quinoa.

### 3.3. Changes in pH, Titratable Acidity (TTA) and Number of Viable Microbial

Both pH and TTA are essential indexes of the fermentation process since these variations are reflected in microorganism metabolism. The pH changes in the fermented quinoa flour under different treatments are illustrated in Figure 4A,B. Regardless of the treatment used, the pH levels of the native, CQ−4, and CQ−20 quinoa flour declined, with a simultaneous rise in TTA contents, after fermentation. This can be ascribed to the production of lactic acid by *L. plantarum* during the fermentation of the quinoa [23]. Another study suggested that the reduction in pH was most likely attributable to the changes of carbohydrates into organic acids by microorganisms present in the quinoa flour [14]. The reduced TTA and elevated pH values were seen in the fermented native samples, which may result from the original low number of LABs causing a prolonged delay period, thus reducing the metabolism and the microbiological viability [4]. During the fermentation of the native quinoa flour, pH values significantly declined, from 6.31 to 3.92. There was a dramatic increase in the number of *L. plantarum* within the first 24 h, after which no further changes were observed. The insignificant change in pH after 24 h can be attributed to the inhibition of microbial growth by the organic acids produced in the fermentation process. This is due to an improved number of protons, which acidified the cytoplasm as well as inhibiting metabolic activities [5]. The pH values of CQ−4 and CQ−20 were similarly reduced, compared to the native quinoa flour. The pH values decreased significantly from 6.31 to 4.51 and 4.17, respectively, in the CQ−4 and CQ−20 samples. This was probably associated with the osmotic pressure of the cell membrane [12]. The pH values continued to decrease as the fermentation time extended, reaching a minimum of 3.95 and 4.05, respectively, in CQ−4 and CQ−20 after 72 h of fermentation. 

The microbial counts of the native, CQ−4, and CQ−20 samples are displayed in Figure 4C. The population of *L. plantarum* increased sharply during the initial 12 h, after which it continued to increase, but much more slowly. It was found that the initial LAB counts in the native quinoa flour were 1.06 CFU/mL, and that they grew dramatically during fermentation, arriving 15.59 CFU/mL after 72 h, and then remained almost static throughout the remainder of the fermentation process. This is probably related to the sustained depletion of nutrients during the first 48 h of fermentation and the limitation of strain growth and reproduction thereafter. Thus, after 48 h, the amount of viable bacteria decreased slightly in the native quinoa flour. However, quinoa flour is a complicated ingredient, containing phenolic compounds, protein, dietary fiber, and some minerals [14]. Hence, other chemical components may be present during fermentation to influence the pH, lactic acid content and cell count. Moreover, cold stress treatment could also enhance the cell count. The higher cell numbers under cold stress treatment were probably due to the fact that the cold stress produced many small pores, broke connections between macromolecules and increased the permeability of the cell membranes [24,25]. However, synergistic treatment had little effect on the bacterial count. 

### 3.4. Enzyme Activity

GAD enzyme activity is responsible for the high GABA production capacity of LAB. Therefore, GAD enzyme activity is an essential indicator of food GABA production [19]. The GAD enzyme activity of the three quinoa samples is illustrated in Figure 3. Their GAD enzyme activity increased gradually at the beginning of fermentation and reached a peak at 48 h. However, GAD enzyme activity was inhibited after 48 h of fermentation time. The initial increase in GAD enzyme activity correlated with the GAD enzyme activity in *L. plantarum*. The initial level of GAD activity was also correlated with that in *L. plantarum*. As the fermentation time increased, the strain activity was affected and the strain growth was considerably inhibited, resulting in a decrease in GAD enzyme activity. The results were consistent with earlier work [26]. The GAD enzyme activity increased under cold stress conditions. The increase in GAD enzyme activity in the quinoa was directly related to the formation of Ca^2+^-calmodulin, leading to GAD activation [12]. A limited but positive effect on GAD activity was shown by the synergistic treatment.

### 3.5. Changes in Crystalline Structure

In agreement with earlier research, the crystalline structure of the quinoa flour was shown to display a typical A-type structure with strong diffraction peaks around 15°, 17°, 18° and 23° (2*θ*) in Figure 5A−C [3]. There was no obvious difference in the position of the strong peaks at different fermentation times. Moreover, neither the fermentation nor the cold stress treatment could significantly affect the position of the above XRD peaks. This indicated that the crystalline structural characteristics of quinoa flour remained unchanged after the cold stress treatment combined with fermentation. The degree of crystallinity in the native quinoa flour was 38.90%. The results showed that fermentation decreased the relative crystallinity. The reduction in crystallinity during the fermentation process was thought to be attributable to the swelling and partial gelatinization of the barley [27]. In our study, the components with large molecular sizes of quinoa were hydrolyzed into smaller molecular sizes within 72 h of fermentation. This increased the swelling and reduced the interaction between the molecular chains, thus reducing the crystallinity. The relative crystallinity was usually correlated with the amount and the average chain length of amylopectin and the moisture content of the starch particles [28]. In the present study, the relative crystallinity of the fermented quinoa flour after the cold stress treatment may have been reduced by the pores on the surface of the quinoa flour (Figure 6), resulting in the erosion of weak crystalline structure. Further disruption of the crystal structure was promoted by the cold-stress-combined fermentation treatment. At a scattering angle (2*θ*) of 20.1°, a distinct Bragg peak may also be noticed, as is typical of the starch-lipid complex [29]. Despite the change in peak shape, a plausible explanation was found for the noticed variations in the starch-lipid complex, which depend on the different treatment conditions, and subsequent investigation by FTIR may contribute to a clearer understanding of these variations. Subsequent studies are consequently required to identify if other factors are responsible for the reduction in crystallinity of the cold stress or fermentation treatment, like the molecular weight and the degree of branching.

### 3.6. Changes in Fourier Transform Infrared Spectroscopy (FTIR)

The deconvoluted FTIR spectra of the native, CQ−4 and CQ−20 samples over 4000−500 cm^−1^ are listed in Figure 5D−F, and the calculated parameters are listed in Table 1. All FTIR spectra were similar, without the appearance of new peaks, indicating that no functional groups changed under the different treatments of the quinoa. The results showed that the effect of fermentation on the peak levels was limited. However, the peaks decreased sharply after fermentation. The FTIR spectra were compared, and the results (Figure 5D–F) demonstrated high absorption peaks at about 3200 cm^−1^, revealing the existence of -OH and polyhydroxy chemicals consisting of phenolic compounds, alcohols, water and other -OH oxygenates [3]. A distinct variation was observed among the peaks at 3200 cm^−1^ due to differences in fermentation times. The trend presented as an increase at 12 h followed by a decrease from 12 h to 48 h in the native quinoa flour, while a gradual decrease from 0−48 h was found in the flours that had undergone the cold stress treatment. This indicates that the -OH bond was broken at different times and that a more violent breakage of the -OH bond was produced by cold stress. Based on the study, the distinctive GABA peak showed -CH peaks at 2841.62 cm^−1^, 2936.23 cm^−1^ and 2954.59 cm^−1^ [30]. Significant differences can be observed among the different fermentation times (Figure 5D−F). The second structure of the protein was represented by an amide I band in the FTIR spectrum (1600−1700 cm^−1^). As can be seen from the small graphs in Figure 5D–F, there was a clear difference in the peaks in the interval of 1600−1700 cm^−1^. Different trends were shown in each of the three different samples. A clear disruption of the protein secondary structure in the native quinoa sample occurred at 12 h, which was probably due to the proteolytic activity of *L. plantarum*. This is the same explanation that Falah et al. reached [31]. In contrast, an enhancement of the protein secondary structure was shown in the CQ−4 and CQ−20 samples at the fermentation time of 12 h, and the disruption of the protein secondary structure only started to become visible at 24 h. It is possible that the start of the destruction of the protein was delayed due to the low temperature.

The absorbance ratio 1047/1022 (R_1047/1022_) is often considered as an indicator of the number of ordered crystalline structures as opposed to the amount of amorphous substance in a starch gel [32]. As shown in Table 1, the intensity ratio of 1047/1022 cm^−1^ for the native, CQ−4 and CQ−20 quinoa ranged from 1.52 to 1.31, and the CQ−20 unfermented quinoa exhibited the highest R_1047/1022_ value (1.52). The trend of the ordered structure in the native quinoa was presented as increasing and then decreasing. However, with the increase of fermentation time, the value of R_1047/1022_ decreased during the cold stress treatment, indicating that the ordered crystalline structures were broken and the destruction of the ordered structure was accelerated during the cold stress treatment. 

### 3.7. Scanning Electron Microscopy (SEM)

Scanning electron microscopy (SEM) images of the quinoa flour are illustrated in Figure 6. In contrast to previous reports, the appearances of the native quinoa flour granules were slightly rough and polyhedral and angular in surface [3]. The structure of the native quinoa flour was intact and cluster shaped, while the cold stressed samples had large, disaggregated particles. This difference in the quinoa treated in the cold stress may be due to cell membrane permeation, and clipped villus and formed holes also can be seen. The observed improvement in the pore numbers could be one of the reasons, due to the intensified hydration of cold-stressed quinoa at the soaking stage [33]. Through increasing the permeability of the quinoa’s surfaces and making more water entry pathways, this change in the surface microstructure accelerated the quinoa’s fermentation. Native unfermented quinoa has a more complete grain structure, whereas the treatment of the CQ−4 and CQ−20 samples disrupts the integrity of the particles and the network structures, resulting in more holes and cracks on the surfaces of the samples. The large quantities of water being combined affected the density of the structured networks, resulting in the disruption of the network formation of the fat and protein, a phenomenon that could be attributed to the cold stress treatment. As mentioned in the previous section, the release of small polysaccharides and free amino acids from the quinoa matrix due to the degradation of macromolecules increased the formation of aggregates on the surface of the samples [25]. As the fermentation time increased, mucus, faults and cracks began to be observed on the surface of the samples.

### 3.8. WAI and WSI

The cold stress treatment is a non-thermal processing method, and thus the WAI and WSI indexes of the quinoa flour samples were measured in the slurry at 25 °C to assess the quinoa flour samples in non-thermal food processing. The statistical analysis of the data is summarized in Table 2, which reveals that fermentation and the cold stress treatment markedly affected WAI and WSI.

The information on the extent of the starch conversion and the release of soluble polysaccharides during the processing was provided by the WSI of the cold stress treated and fermented samples. An increase in WAI and WSI was manifested by CQ−4 and CQ−20 compared to the native quinoa flour, which could be attributed to the cold stress resulting in increased soluble sugars and free amino acids [34]. The value of the WSI increased remarkably within 48 h of lactic acid fermentation. In fermentation, this increase was attributed to the decrease in pH causing the denaturation of some of the proteins and generating other insoluble compounds [35].

The WAI value was influenced by the number of hydrophilic groups and the gel formation capacity [17]. A higher WAI was exhibited in the CQ−4 and CQ−20 samples than in the native quinoa flour. This could be due to a disassembly of polysaccharide molecules, a change in the amount of protein and an increase in the protein level, resulting in increased opportunities to interact with and hold water [36]. The same increasing trend within 48 h of fermentation was observed both with and without the cold stress treatment. These changes were probably due to partial protein denaturing and starch damage. In addition, the hydrolysis of biological macromolecules like proteins and starches into smaller molecules by the proteases and amylases of microorganisms may also contribute to these variations [37].

### 3.9. Pasting Properties 

The differences in the pasting and gelling behaviors of the cold-stressed and fermented quinoa flours were shown for comparison (Table 3). The peak viscosity of the native quinoa flour was 774 cP, which decreases markedly to 617 cP and 486 cP after the application of the 4 °C and −20 °C cold stress treatments, respectively. During 36 h of fermentation with or without cold stress, dramatic losses of PV, TV, BV, FV, SV and PT were detected. A notable drop in viscosity values resulted after the 36 h of fermentation time. Nonetheless, the trend in all of the RVA parameters for Native, CQ−4 and CQ−20 was generally consistent. However, the variations of these parameters in CQ−4 and CQ−20 were significantly reduced compared to the native quinoa flour. The peak viscosity related to the viscosity of the starch granules when completely gelatinized, which can be connected to the expanding capacity of the starch granules [38]. The decreased values of CQ−4 and CQ−20 indicate that the fermented quinoa flour tended to expand at colder temperatures (Table 2). 

One of the most important factors affecting the swelling of starch granules is the content of the straight-chain starch, whereas the swelling of starch granules was inhibited by the presence of lipids and amylose in grain [2]. Straight-chain starch was more soluble at higher freezing temperatures, so the peak increased. Meanwhile, the pasting properties of quinoa flour may be influenced by amylase activity, protein content and starch content [2]. Thus, the decrease in the PV, TV, BV, FV and SV of the fermented quinoa flours may be due to amylase and protease activation, which causes starch and protein breakdown.

A lower peak viscosity was found in the cold-stress-treated samples than in the native quinoa flour, suggesting that the cold stress treatment could decrease the peak viscosity of quinoa starch. The reduction in the peak, breakdown, final and setback viscosities and the pasting temperature of the quinoa flours after fermentation may be due to starch degradation. As the freezing temperature decreased, the gelation temperature of the quinoa flour increased, probably due to the increased crystallinity of the starch granules and insufficient energy to destroy the stable structure at colder temperatures as discussed by Su et al. [38], though this result was contrary to our study (Table 1). The peak viscosity of the starch decreased because the starch polymer degraded, the molecular structure changed, and the binding force between the starch chains decreased as a result of the cold treatment [39].

## 4. Conclusions

*L. plantarum* is one of the most typical and common prebiotic bacteria found in fermented foods. To maximize the amount of GABA in fermented quinoa, which has practical applications for the food industry, we designed a synthetic method of cold stress combined with fermentation. The largest GABA production amount and best GAD enzyme activity were found in *Lactobacillus plantarum* NO.2, out of sixteen strains tested. The level of GABA in the fermented quinoa was elevated by 1191% and 774% after cold stress treatments at 4 °C and −20 °C, respectively, compared with untreated quinoa. More holes, cracks and fractures were shown in the morphology of the fermented quinoa flour after cold stress, which was due to the expansion produced by the low temperature. After fermentation, in contrast to the native quinoa, the secondary structures of proteins were disrupted at a slower rate in the cold-stress-treated quinoa, as revealed by the analysis on FTIR spectra. Clear differences were also shown in the GABA typical -CH bond at different fermentation times. In conclusion, the study results can serve as a reference for the future manipulation of GABA-enriched functional quinoa foods production on an industrial scale.

## Figures and Tables

**Figure 1 foods-11-03908-f001:**
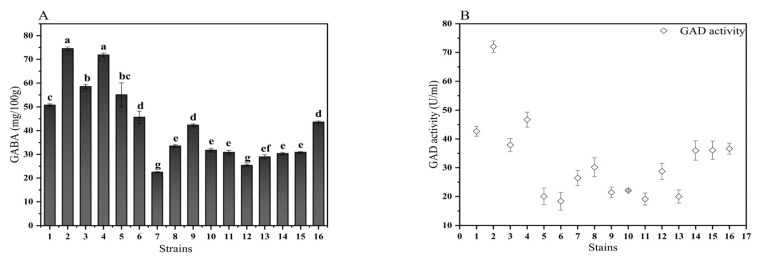
GABA contents of the culture filtrates of the isolates grown in quinoa medium (**A**). Change in GAD activity using different *L. plantarum*, NO.1-NO.16 (**B**). GABA: γ-aminobutyric acid; Native: native quinoa flour; CQ−4: quinoa flour cold stress treated at 4 °C; CQ−20: quinoa flour cold stress treated at −20 °C. Results are expressed as the mean ± standard deviation. Different lowercase letters indicate significant differences (*p* < 0.05) between different strains.

**Figure 2 foods-11-03908-f002:**
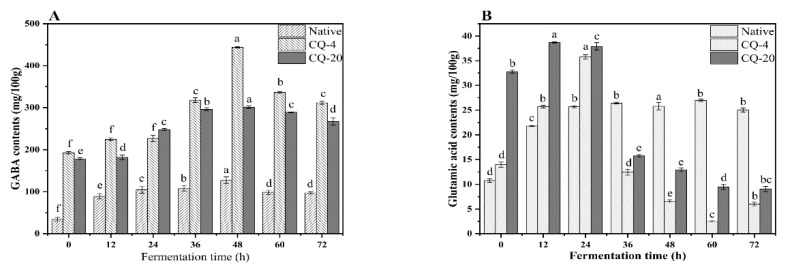
Effect of fermentation with cold stress at 4 °C and −20 °C on the GABA contents (**A**) and glutamic acid contents (**B**). GABA: γ-aminobutyric acid; Native: native quinoa flour; CQ−4: quinoa flour cold stress treated at 4 °C; CQ−20: quinoa flour cold stress treated at −20 °C. Different lowercase letters indicate significant differences (*p* < 0.05) between the same sample group (Native, CQ−4, CQ−20).

**Figure 3 foods-11-03908-f003:**
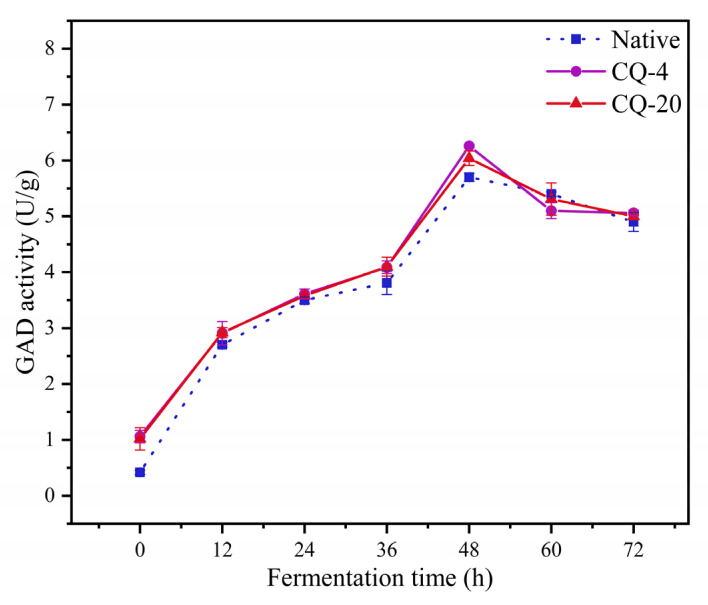
Variation of the GAD activity content during the different fermentation processing. Native: native quinoa flour; CQ−4: quinoa flour cold stress treated at 4 °C; CQ−20: quinoa flour cold stress treated at −20 °C. Values are reported as mean ± standard deviation. GAD: glutamate decarboxylase.

**Figure 4 foods-11-03908-f004:**
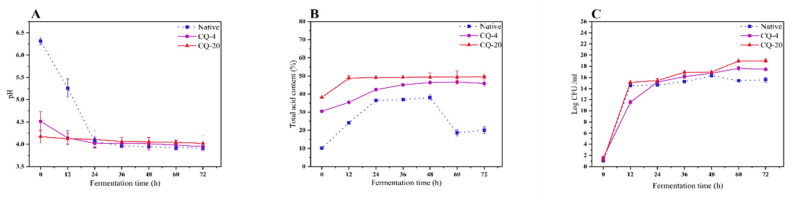
Fermentation time-dependence of pH (**A**), total acid content (**B**), and viable cell count (**C**) in Native, CQ−4, CQ−20 samples fermented by *Lactobacillus plantarum* NO.2 at 34 °C. Native: native quinoa flour; CQ−4: quinoa flour cold stress treated at 4 °C; CQ−20: quinoa flour cold stress treated at −20 °C. Values are reported as mean ± standard deviation.

**Figure 5 foods-11-03908-f005:**
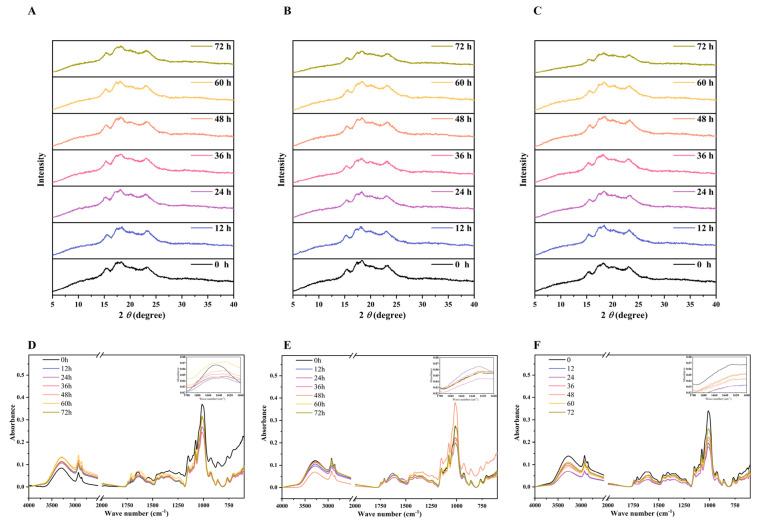
X-ray diffraction pattern (**A**–**C**) and FTIR absorbance spectrum (**D**–**F**) of native and fermented quinoa flour under cold stress at 4 °C and −20 °C. FTIR: Fourier transform infrared spectroscopy; Native: native quinoa flour; CQ−4: quinoa flour cold stress treated at 4 °C; CQ−20: quinoa flour cold stress treated at −20 °C.

**Figure 6 foods-11-03908-f006:**
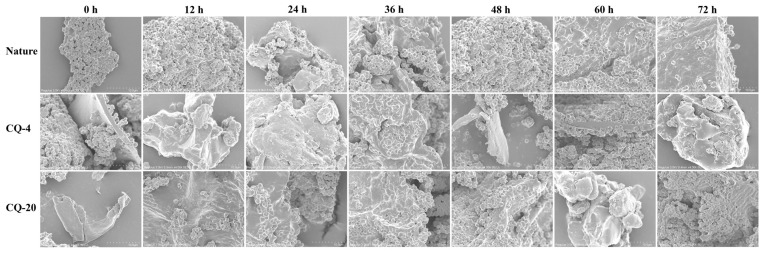
The scanning electron microscopy (SEM) of native and fermented quinoa flour under cold stress at 4 °C and −20 °C. Native: native quinoa flour; CQ−4: quinoa flour cold stress treated at 4 °C; CQ−20: quinoa flour cold stress treated at −20 °C. The letter A−U represents different samples.

**Table 1 foods-11-03908-t001:** The XRD patterns, FTIR intensity ratio and soluble sugar of native and fermented quinoa flour after cold stress.

Samples		Degree of Crystallinity (%)	Crystal Type	R_1047/1022_	Soluble Sugar (%)
Groups	Fermentation Time (h)				
Native	0	38.89 ± 0.78 ^a^	A	1.31 ± 0.01 ^b^	4.48 ± 0.02 ^b^
	12	11.68 ± 0.31 ^ab^	A	1.51 ± 0.01 ^bc^	2.41 ± 0.10 ^b^
	24	9.58 ± 0.72 ^b^	A	1.55 ± 0.01 ^c^	1.79 ± 0.07 ^d^
	36	7.48 ± 0.71 ^b^	A	1.46 ± 0.03 ^c^	0.76 ± 0.01 ^d^
	48	7.03 ± 0.41 ^a^	A	1.46 ± 0.04 ^a^	0.25 ± 0.03 ^a^
	60	6.97 ± 0.38 ^bc^	A	1.43 ± 0.02 ^cd^	0.24 ± 0.02 ^c^
	72	6.71 ± 0.19 ^c^	A	1.42 ± 0.01 ^c^	0.23 ± 0.02 ^c^
CQ−4	0	10.6 ± 0.51 ^a^	A	1.45 ± 0.01 ^ab^	6.47 ± 0.04 ^d^
	12	9.06 ± 0.81 ^bc^	A	1.44 ± 0.01 ^b^	4.73 ± 0.09 ^c^
	24	8.94 ± 0.79 ^c^	A	1.41 ± 0.04 ^b^	3.56 ± 0.07 ^c^
	36	7.02 ± 0.01 ^d^	A	1.39 ± 0.03 ^a^	3.01 ± 0.02 ^b^
	48	6.83 ± 0.84 ^ab^	A	1.36 ± 0.01 ^a^	1.25 ± 0.03 ^b^
	60	6.82 ± 0.57 ^c^	A	1.35 ± 0.02 ^b^	0.79 ± 0.02 ^a^
	72	6.15 ± 0.67 ^b^	A	1.36 ± 0.01 ^c^	0.33 ± 0.04 ^cd^
CQ−20	0	8.61 ± 0.01 ^a^	A	1.52 ± 0.01 ^d^	6.33 ± 0.06 ^c^
	12	7.95 ± 1.00 ^b^	A	1.48 ± 0.02 ^b^	4.26 ± 0.05 ^c^
	24	7.59 ± 0.83 ^bc^	A	1.46 ± 0.01 ^c^	3.79 ± 0.03 ^d^
	36	7.23 ± 0.62 ^a^	A	1.43 ± 0.01 ^b^	2.05 ± 0.04 ^b^
	48	6.83 ± 0.21 ^a^	A	1.43 ± 0.01 ^ab^	1.31 ± 0.02 ^a^
	60	6.80 ± 0.01 ^c^	A	1.43 ± 0.04 ^c^	0.31 ± 0.06 ^a^
	72	6.34 ± 0.39 ^b^	A	1.45 ± 0.02 ^d^	0.29 ± 0.09 ^b^

Native: native quinoa flour; CQ−4: quinoa flour cold stress treated at 4 °C; CQ−20: quinoa flour cold stress treated at −20 °C. XRD: X-ray diffraction; FTIR: Fourier transform infrared spectroscopy. Values followed by different letters on the same column indicate statistical differences between the same sample group for the same parameter (*p* < 0.05).

**Table 2 foods-11-03908-t002:** Effect of different fermentation time after cold stress processes on the WAI and WSI.

Fermentation Times (h)	WSI (g/g)			WAI (g/100g)		
	Native	CQ−4	CQ−20	Native	CQ−4	CQ−20
0	16.33 ± 0.01 ^d^	21.03 ± 0.01 ^d^	27.85 ± 0.02 ^ab^	2.13 ± 0.35 ^b^	2.54 ± 0.22 ^d^	2.40 ± 0.05 ^b^
12	19.66 ± 0.01 ^c^	22.21 ± 0.03 ^d^	28.20 ± 0.02 ^ab^	2.50 ± 0.22 ^a^	2.59 ± 0.17 ^d^	2.46 ± 0.02 ^b^
24	22.66 ± 0.02 ^a^	23.34 ± 0.01 ^d^	29.94 ± 0.03 ^a^	2.50 ± 0.20 ^a^	2.59 ± 0.15 ^d^	2.58 ± 0.03 ^ab^
36	22.92 ± 0.05 ^a^	23.99 ± 0.01 ^d^	28.62 ± 0.02 ^ab^	2.51 ± 0.26 ^a^	2.63 ± 0.33 ^c^	2.65 ± 0.02 ^ab^
48	23.32 ± 0.01 ^b^	27.48 ± 0.05 ^b^	30.59 ± 0.01 ^a^	2.58 ± 0.15 ^a^	2.65 ± 0.13 ^bc^	2.73 ± 0.07 ^a^
60	20.71 ± 0.02 ^b^	25.94 ± 0.03 ^bc^	26.28 ± 0.03 ^bc^	2.51 ± 0.09 ^a^	2.59 ± 0.04 ^d^	2.57 ± 0.04 ^ab^
72	20.08 ± 0.01 ^b^	25.40 ± 0.02 ^bc^	21.81 ± 0.09 ^c^	2.51 ± 0.19 ^a^	2.52 ± 0.10 ^d^	2.35 ± 0.27 ^b^

Native: native quinoa flour; CQ−4: quinoa flour cold stress treated at 4 °C; CQ−20: quinoa flour cold stress treated at −20 °C.WAI: water absorption index WSI: water solubility index. Values followed by different letters on the same column indicate statistical differences between the same sample group for the same parameter (*p* < 0.05).

**Table 3 foods-11-03908-t003:** Pasting properties of native and fermented quinoa under 4 °C and −20 °C cold stress conditions.

Varieties	Fermentation Time (h)	PV (cP)	BV (cP)	TV (cP)	SV (cP)	FV (cP)	PT (°C)
Native	0	774 ± 4.9 ^d^	107 ± 15.8 ^c^	667 ± 9.1 ^a^	129 ± 10.6 ^bc^	796 ± 12.2 ^b^	94.90 ± 13.7 ^d^
	12	683 ± 5.1 ^a^	85 ± 14.6 ^a^	598 ± 5.9 ^b^	217 ± 14.0 ^a^	815 ± 13.1 ^b^	94.80 ± 15.7 ^a^
	24	606 ± 9.1 ^a^	65 ± 14.1 ^a^	567 ± 13.4 ^c^	190 ± 3.9 ^a^	757 ± 5.6 ^a^	51.20 ± 13.4 ^a^
	36	604 ± 14.7 ^a^	37 ± 14.5 ^ab^	541 ± 2.6 ^d^	190 ± 10.7 ^a^	731 ± 21.3 ^b^	50.55 ± 12.5 ^a^
	48	761 ± 2.9 ^a^	73 ± 24.6 ^c^	688 ± 21.0 ^bc^	212 ± 9.0 ^b^	900 ± 14.6 ^b^	56.30 ± 10.9 ^a^
	60	674 ± 8.7 ^a^	46 ± 12.0 ^b^	628 ± 7.0 ^a^	168 ± 19.4 ^b^	796 ± 9.1 ^c^	51.10 ± 9.8 ^c^
	72	719 ± 11.5 ^b^	103 ± 17.9 ^ab^	616 ± 12.1 ^b^	213 ± 10.0 ^e^	829 ± 6.7 ^d^	59.05 ± 6.9 ^c^
CQ−4	0	617 ± 8.8 ^b^	154 ± 11.6 ^b^	463 ± 15.7 ^b^	254 ± 24.7 ^a^	717 ± 21.7 ^d^	95.35 ± 9.3 ^a^
	12	546 ± 8.5 ^a^	106 ± 20.6 ^a^	432 ± 21.5 ^a^	178 ± 10.5 ^bc^	618 ± 12.3 ^d^	50.35 ± 10.3 ^a^
	24	544 ± 9.3 ^a^	101 ± 20.1 ^b^	431 ± 24.1 ^a^	169 ± 5.1 ^b^	600 ± 21.0 ^c^	50.05 ± 13.7 ^c^
	36	524 ± 7.9 ^c^	93 ± 19.5 ^a^	440 ± 15.8 ^c^	149 ± 6.0 ^a^	581 ± 16.9 ^a^	50.00 ± 19.1 ^b^
	48	608 ± 11.5 ^a^	143 ± 10.5 ^a^	465 ± 18.0 ^a^	241 ± 13.0 ^b^	706 ± 14.2 ^ab^	50.40 ± 6.9 ^b^
	60	564 ± 10.6 ^a^	123 ± 10.4 ^b^	441 ± 8.0 ^c^	179 ± 21.7 ^c^	620 ± 14.0 ^d^	50.10 ± 13.0 ^b^
	72	500 ± 20.8 ^a^	109 ± 9.9 ^c^	391 ± 13.7 ^b^	133 ± 12.9 ^d^	524 ± 15.7 ^c^	51.10 ± 7.3 ^d^
CQ−20	0	494 ± 23.7 ^b^	81 ± 12.6 ^c^	434 ± 25.6 ^a^	157 ± 25.1 ^e^	591 ± 24.6 ^e^	95.35 ± 2.9 ^c^
	12	486 ± 23.4 ^b^	60 ± 10.5 ^b^	405 ± 17.9 ^a^	130 ± 18.9 ^b^	535 ± 26.9 ^b^	50.10 ± 3.9 ^a^
	24	423 ± 8.3 ^b^	42 ± 11.5 ^bc^	387 ± 15.1 ^a^	135 ± 16.4 ^a^	522 ± 14.9 ^c^	50.05 ± 11.0 ^a^
	36	418 ± 5.9 ^a^	36 ± 5.9 ^a^	356 ± 22.3 ^a^	111 ± 16.7 ^c^	487 ± 21.5 ^c^	50.05 ± 5.7 ^a^
	48	454 ± 25.6 ^b^	63 ± 4.6 ^b^	391 ± 15.3 ^b^	146 ± 21.0 ^c^	537 ± 23.3 ^b^	55.10 ± 16.7 ^a^
	60	413 ± 27.9 ^b^	58 ± 20.5 ^b^	352 ± 9.8 ^b^	114 ± 12.5 ^b^	440 ± 21.0 ^a^	50.15 ± 20.9 ^c^
	72	408 ± 30.1 ^a^	56 ± 3.3 ^b^	347 ± 11.3 ^b^	993 ± 19.8 ^b^	466 ± 19.8 ^b^	50.40 ± 25.1 ^b^

PV: peak viscosity; TV: trough viscosity; BD: breakdown; FV: final viscosity; SB: setback; PT: pasting temperature. Native: native quinoa flour; CQ−4: quinoa flour cold stress treated at 4 °C; CQ−20: quinoa flour cold stress treated at −20 °C. All values of factors are expressed as mean ± SD of triplicate experiments. Values followed by different letters in the same column indicate statistical differences between the same sample groups for the same parameter (*p* < 0.05).

## Data Availability

Data is contained within the article.

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
