# Peer review of "Enhance Production of γ-Aminobutyric Acid (GABA) and Improve the Function of Fermented Quinoa by Cold Stress"

_foods, 2022, doi:10.3390/foods11233908_

Round 1

Reviewer 1 Report

I have reviewed the manuscript and following points should be considered, in my opinion:

Ø  The subject of the study is original.

Ø  The wiriting of L.Plantarum is erroneous. It should be written as “L. plantarum” and , if possible, the name of this species might be written according to the new classification.

Ø  What do you mean by 4%? On what medium were the strains grown? Has it been counted? Were cells added or inoculated with the medium? In short, inoculation conditions should be written.

Ø  Line 168: The full name of the abbreviation SEC must be given.

Ø  Statistical analyzes should be reviewed. Two factors are mentioned, namely fermentation time and treatment. However, these factors were not included in the statistical analysis. How many times was the experiment repeated with different raw material? Or were the same samples analyzed 3 times? In the conclusion and discussion part, it is not stated whether the interactions are important or not. Focus should be on the interactions that are considered important rather than the main effects. Was one way or two way used in statistical analysis?

Ø  Multiple comparison test results are not given in Fig.1. How were the differences evaluated?

Ø  Line 202-203: Which one is the correct expression; hours or h?

Ø  Numbering of figures must be checked. Figure 3 must be placed before Figure 4.

Ø  Statistical evaluations in Tables 1, 2 and 3 are incomprehensible. Lettering should be reviewed. Footnotes should be descriptive (column or row)

Ø  Whether the intractions are significant (Fermentation time x treatment) should be highlighted.

Ø  The necessary attention and care should be considered in the writing of the manuscript.

Author Response

Dear reviewer,

Thank you very much for your kind letter, along with the constructive comments of the editor and reviewers concerning our manuscript (ID: foods-2056001). We have thoroughly considered all the comments and substantially revised our manuscript.

The point-to-point answers and explanations for all revisions were listed in a separate paper following this letter.

We have tried our best to address all the concerns raised by the reviewers. We hope, with these modifications and improvements based on the comments of the editor and reviewers, the quality of our manuscript would meet the publication standard of Foods.

Yours sincerely,

Li Wang

___________________________

Dr. Li Wang

State Key Laboratory of Food Science and Technology

National Engineering Research Center of Cereal Fermentation and Food Biomanufacturing

School of Food Science and Technology

Jiangnan University.

Wuxi 214122

  1. R. China

Response to Reviewer

Q1. The subject of the study is original.

A: Thank you for your comments.

Q2. The wiriting of L.Plantarum is erroneous. It should be written as “L. plantarum” and , if possible, the name of this species might be written according to the new classification.

A: Thank you very much for the suggestion. “L.Plantarum” has been modified to “L. plantarum” according to your suggestion.

This strain is derived from the intestinal flora of the elderly in longevity village of Guangxi Province. At present, only 16sDNA sequence determination has been conducted for this strain, and it is identified as Lactobacillus plantarum. Therefore, the name of this species was written according to the new classification.

Q3. What do you mean by 4%? On what medium were the strains grown? Has it been counted? Were cells added or inoculated with the medium? In short, inoculation conditions should be written.

A: Thank you for the suggestion.

4% means adding 100 μL of the bacterial suspension to 2.5 g of quinoa suspensions. MSG medium and quinoa medium can be used for the growth of bacteria. MSG medium was initially used to activate the strain. The quinoa medium is designed for more GABA production by the bacterium. After the cells were activated by MSG medium, they were added to quinoa medium, where they were cultured.

The explanation has been added in the manuscript:

  1. plantarum was initially fostered in MRS Medium at 37 ℃ for 12 h in order to activate the strain and then 100 μL spore suspension was moved from to quinoa substrates (2.50 g) in an attempt to have produced higher GABA. (Line 78-79)

Q4. Line 168: The full name of the abbreviation SEC must be given.

A: Thank you for the suggestion.

We intended to write SEM instead of SEC, but it didn't feel right. So the section has been amended as follows

All experiments were repeated three times and data are presented as the mean of three samples ± standard deviation (SD). (Line 146)

Q5. Statistical analyzes should be reviewed. Two factors are mentioned, namely fermentation time and treatment. However, these factors were not included in the statistical analysis. How many times was the experiment repeated with different raw material? Or were the same samples analyzed 3 times? In the conclusion and discussion part, it is not stated whether the interactions are important or not. Focus should be on the interactions that are considered important rather than the main effects. Was one way or two way used in statistical analysis?

A: Thank you for the suggestion.

In this experiment, three different raw materials were used for sample preparation, and the final results were obtained after three or more tests.

The interaction between fermentation and cold stress is very important. In the conclusion and discussion part, the interaction between fermentation time and cold stress treatment is elaborated in the manuscript as follows:

(1) The maximum GABA boost in quinoa was 26.9% if the fermentation treatment alone was carried out. (Line 173-175)

(2) If only the cold stress treatment was applied, the GABA content in 4 °C and 20 °C cold stress treated quinoa was elevated by 460% and 418% respectively. (Line 185-186)

(3) However, cold stress combined with fermentation treatment was carried out and the maximum elevated GABA content in cold stress fermented quinoa reached 1191%, so this synergistic effect is very important. (Line 193-196) In addition, we have added some effects of the combined treatment on other indicators. For example, bacterial count (Line 243), GAD activity (Line 254-255), crystal structure (Line 270-271).

In this study, one way was used for statistical analysis. (Line 148-149)

Q6. Multiple comparison test results are not given in Fig.1. How were the differences evaluated?

A: Thank you for the suggestion. The experiment has been carried out many times, and the results of multiple comparisons have been shown in Fig. 1, and the differences evaluated have also been marked in Fig. 1.

Q7. Line 202-203: Which one is the correct expression; hours or h?

A: Thank you for the suggestion. “hours” and “h” are both expression units of time, but we should express them in the same way, so we changed the two existing “hours” in the manuscript to “h”.

Q8. Numbering of figures must be checked. Figure 3 must be placed before Figure 4.

A: Thank you for the suggestion. We have placed Figure 4 behind Figure 3 and modified the manuscript accordingly.

Q9. Statistical evaluations in Tables 1, 2 and 3 are incomprehensible. Lettering should be reviewed. Footnotes should be descriptive (column or row)

A: Thank you for the suggestion. Statistical evaluations in Table 1,2,3, we restated as follows:

(Line 305-306, 341-342, 370-372)

Values followed by different letters on the same column indicate statistical differences between the same sample group for the same parameter (p < 0.05).

Q10. Whether the intractions are significant (Fermentation time x treatment) should be highlighted.

A: Thank you for the suggestion.

Cold stress combined with fermentation treatment is very significant. Cold stress combined with fermentation treatment was carried out and the maximum elevated GABA content in cold stress fermented quinoa reached 1191%. For 26.9% of fermentation alone and 460% of cold stress alone, there was a significant improvement effect. Therefore, this synergistic effect is very important.

(Line 193-196)

Q11. The necessary attention and care should be considered in the writing of the manuscript.

A: Thank you very much for the suggestion. We reviewed and carefully revised the entire article in order to minimize writing problems in the manuscript. In addition, the manuscript has been refined by English native speaker.

Reviewer 2 Report

This is an interesting work, only minor changes are suggested; please see below:

Abstract and text. Define non-common abbreviations the first time they appear.

English language. In some few statements readability needs improvement. And some few grammar mistakes are found in text.

Introduction L. 77-78. "...cereal products"? You stated at the beginning that quinoa is a pseudocereal, and that is right. Please correct. 

Section 2.2. Indicate clearly the meaning of 4% (inoculum)

 Figure 1 and remaining figures and tables. Define in the figure caption, and at the bottom of tables, definitions of the corresponding non-common abbreviations.

References. Some journal titles are abbreviated, some not; the same journal title is abbreviated and not; some references are wrongly cited and/or incomplete. Please check all of them.

At the end, I would like to congratulate you for this interesting research work. 

Author Response

Dear reviewer,

Thank you very much for your kind letter, along with the constructive comments of the editor and reviewers concerning our manuscript (ID: foods-2056001). We have thoroughly considered all the comments and substantially revised our manuscript.

The point-to-point answers and explanations for all revisions were listed in a separate paper following this letter.

We have tried our best to address all the concerns raised by the reviewers. We hope, with these modifications and improvements based on the comments of the editor and reviewers, the quality of our manuscript would meet the publication standard of Foods.

Yours sincerely,

Li Wang

___________________________

Dr Li Wang

State Key Laboratory of Food Science and Technology

National Engineering Laboratory of Cereal Fermentation Technology

School of Food Science and Technology

Jiangnan University.

Wuxi 214122

  1. R. China

Response to Reviewer

Q1. Abstract and text. Define non-common abbreviations the first time they appear.

A: Thanks for your suggestion. We defined the abbreviations the first time they appeared. For example: “GABA” has been defined to “γ-aminobutyric acid (GABA)” (Line 15); “L. plantarum” has been defined to “Lactobacillus plantarum” (Line 17); “FTIR” has been corrected to “Fourier-transform infrared spectrophotometer (FTIR)” (Line 21); “RVA” has been corrected to “rapid visco analyzer (RVA)” (Line 23);

Q2. English language. In some few statements readability needs improvement. And some few grammar mistakes are found in text.

Thanks for your suggestion. We have improved the language of the manuscript according to comments. And some grammar mistakes were corrected. For example, “has” has been revised to “have” (Line 59).

Introduction L. 77-78. "...cereal products"? You stated at the beginning that quinoa is a pseudocereal, and that is right. Please correct. 

A: Thanks for your suggestion. Introduction L. 70-71. "...cereal products" have been modified to be " pseudocereal products".

Q3. Section 2.2. Indicate clearly the meaning of 4% (inoculum)

A: Thank you for the suggestion. 4% means adding 100 μL of the bacterial suspension to 2.5 g of quinoa suspensions.

The explanation has been added in the manuscript:

  1. plantarum was initially fostered in MRS Medium at 37 ℃ for 12 h in order to activate the strain and then 100 μL spore suspension was moved from to quinoa substrates (2.50 g) in an attempt to have produced higher GABA. (Line 78-79)

Q4. Figure 1 and remaining figures and tables. Define in the figure caption, and at the bottom of tables, definitions of the corresponding non-common abbreviations.

A: The corresponding non-common abbreviations have been defined at the bottom of the graph and table. The changes have been highlighted in red. For example, the full name of “GABA” (Line 168, 203); “GAD” (Line 232); “XRD”, “FTIR” (Line 305); “SEM” (Line 331) has been added to the article.

Q5. References. Some journal titles are abbreviated, some not; the same journal title is abbreviated and not; some references are wrongly cited and/or incomplete. Please check all of them.

A: We have checked the format of all references, and all journal titles have adopted non-abbreviations references and removed wrongly cited and/or incomplete references. These changes have been highlighted in red.

Round 2

Reviewer 1 Report

It can be accepted for the publication.